# Induction of Antimicrobial Protein S100A15 Expression by Oral Microbial Pathogens Is Toll-like Receptors-Dependent Activation of c-Jun-N-Terminal Kinase (JNK), p38, and NF-κB Pathways

**DOI:** 10.3390/ijms24065348

**Published:** 2023-03-10

**Authors:** Denis Selimovic, Naji Kharouf, Florence Carrouel, Sofie-Yasmin Hassan, Thomas W. Flanagan, Sarah-Lilly Hassan, Mosaad Megahed, Youssef Haikel, Simeon Santourlidis, Mohamed Hassan

**Affiliations:** 1Institut National de la Santé et de la Recherche Médicale, University of Strasbourg, 67000 Strasbourg, France; 2Department of Restorative Dentistry, Endodontology and Biomaterials, Faculty of Dentistry, University of Tours, 37000 Tours, France; 3Department of Operative Dentistry and Endodontics, Dental Faculty, University of Strasbourg, 67000 Strasbourg, France; 4Health, Systemic, Process, UR 4129 Research Unit, University Claude Bernard Lyon 1, University of Lyon, 69008 Lyon, France; 5Department of Chemistry, Faculty of Science, Heinrich-Heine University Duesseldorf, 40225 Duesseldorf, Germany; 6Department of Pharmacology and Experimental Therapeutics, LSU Health Sciences Center, New Orleans, LA 70112, USA; 7Clinic of Dermatology, University Hospital of Aachen, 52074 Aachen, Germany; 8Institute of Cell Therapeutics and Diagnostics, University Medical Center of Duesseldorf, 40225 Duesseldorf, Germany; 9Research Laboratory of Surgery-Oncology, Department of Surgery, Tulane University School of Medicine, New Orleans, LA 70112, USA

**Keywords:** S100A15, TLR2, TLR4, ASK1, JNK, p38, LPS, LTA, NF-κB

## Abstract

The antimicrobial protein S100A15 belongs to the S100 family, which is differentially expressed in a variety of normal and pathological tissues. Although the function of S100A15 protein has been discussed in several studies, its induction and regulation in oral mucosa, so far, are largely unknown. In this study, we demonstrate that S100A15 is induced by the stimulation of oral mucosa with gram^−^ or gram^+^ bacterial pathogens, as well as with the purified membrane components, namely lipopolysaccharides (LPS) and lipoteichoic acid (LTA). The stimulation of the human gingival fibroblast (GF) and the human mouth epidermal carcinoma (KB) cell lines with either gram^−^ or gram^+^ bacterial pathogens or their purified membrane components (LPS and LTA) results in the activation of NF-κB, apoptosis-regulating kinase1 (ASK1), and MAP kinase signaling pathways including, c-Jun N-terminal kinase (JNK) and p38 together with their physiological substrates AP-1 and ATF-2, respectively. Inhibition of S100A15 by antibodies-mediated Toll-like receptor 4 (TLR4) or Toll-like receptor 2 (TLR2) neutralization reveals the induction of S100A15 protein by LPS/gram^−^ bacterial pathogens to be TLR4- dependent mechanism, whereas induction by LTA/gram^+^ bacterial pathogens to be TLR2- dependent mechanism. Pre-treatment of GF and KB cells with JNK (SP600125), p38 (SB-203580), or NF-κB (Bay11-7082) specific inhibitors further demonstrates the importance of JNK, p38 and NF-κB pathways in the regulation of gram^−^/gram^+^ bacterial pathogen-induced S100A15 expression. Our data provide evidence that S100A15 is induced in cancer and non-cancer oral mucosa-derived cell lines by gram^−^/gram^+^ bacterial pathogens and provide insight into the molecular mechanisms by which gram^−^ and gram^+^ bacterial pathogens induce S100A15 expression in the oral mucosa.

## 1. Introduction

The S100 family of calcium-binding proteins comprises more than 20 members that are differentially expressed in a variety of both normal and abnormal tissues [1,2,3]. Due to their wide expression profile, S100 proteins are largely characterized by their functional diversification [4,5]. The biological activities of S100 proteins vary widely and include calcium-dependent roles in inflammation and regulation of cellular processes, including protein phosphorylation, cell proliferation, structural membrane, and cytoskeletal organization [6,7]. S100 expression plays putative functional roles in epidermal cell maturation, tumorigenesis, inflammation, and adaptative immunity [8,9,10,11]. S100 proteins are also involved in the modulation of a wide variety of intra- and extracellular functions, including cytokine-like and chemokine-like modulation of advanced glycation end products (RAGE) and Toll-like receptor 4 (TLR4) dependent signaling cascades [12]. S100A15 protein, in particular, is infamous for its antimicrobial property in vitro and in vivo [13]. Although the induction of S100A15 in normal human keratinocytes by Lipopolysaccharides (LPS) or *E. coli* stimulation has been reported [13], its induction and regulation in human oral mucosa, so far, is unknown.

Although the epithelial tissues in the oral cavity are constantly exposed to a variety of bacteria, most individuals maintain healthy homeostasis. This non-pathogenic homeostasis is attributed to the stratified squamous epithelium of the oral cavity, which functions as a barrier between the host and outside environment, as well as the innate host response, i.e., expression of antimicrobial peptides via epithelial cells [14]. These antimicrobial peptides have a broad spectrum of activity against both Gram-negative and Gram-positive bacteria as well as yeast and viruses [15,16,17,18,19,20].

In this study, we evaluated if the expression of the antimicrobial protein S100A15 is inducible by either gram^−^- or/and gram^+^ bacterial pathogens in oral mucosa, and if so, what molecular mechanisms govern this expression. Ultimately, we demonstrate that the induction of the antimicrobial protein S100A15 in cancer and non-cancer oral mucosa-derived cell lines are mediated by TLR4/2-dependent mechanisms.

## 2. Results

### 2.1. Induction of the Antimicrobial Protein S100A15 by LPS, LTA, and Bacterial Pathogens in Both GF and KB Cells

To demonstrate whether stimulation of GF and KB cells with either heat-inactivated gram^−^ (*P. gingivalis*) or heat-inactivated gram^+^ (*S. sanguinis*) results in the induction of S100A15 expression, GF and KB cells were stimulated with lipopolysaccharides (LPS), heat-inactivated *P. gingivalis*, lipoteichoic acid (LTA) or heat-inactivated *S. sanguinis* for varying time intervals up to 72 h, then, total RNA was extracted, and the RT-PCR performed. The analysis of the mRNA level of S100A15 (Figure 1) demonstrates the induction of S100A15 in both GF (Figure 1A) and KB (Figure 1B) cells by stimulation with LTA (50 ng/mL), as well as with LPS (100 ng/mL) stimulation (Figure 1C,D). S100A15 expression was first noted at 6 h following either LTA (Figure 1A,B) or LPS (Figure 1C,D) stimulation. Although the noted expression level of S100A15 by the stimulation with LTA and LPS is quite different, the induced expression of S100A15 in Both GF and KB cells following the stimulation with either LTA or LPS increased thereafter at 24h to reach the maximum at 48 h without any alteration at 72 h, suggesting that the stimulation of both cell lines with either LTA or LPS for 48 h is the optimal time point at which the expression of S100A15 protein can be induced in oral mucosa-derived cells. To confirm whether the LTA or LPS-induced expression of S100A15 can be induced by the stimulation of the same cell lines with either heat-inactivated gram^−^ (*P. gingivalis*) or gram^+^ (*S. sanguinis*) bacterial pathogens. We stimulated both GF and KB cells with heat-inactivated *P. gingivalis* (2 × 10^4^ cells/mL) or heat-inactivated *S. sanguinis* (2 × 10^4^ cells/mL) for a regulated time interval of up to 72 h. As expected, the expression of S100A15 was noted first in both GF and KB cells 6 h following the stimulation with either heat-inactivated *P. gingivalis* (Figure 1E) or heat-inactivated *S. sanguinis* (Figure 1F) and increased thereafter to reach the maximum at 48 h post-stimulation. In addition, we analyzed the induced expressions of S100A15 protein in both GF and KB cells after the stimulation with either heat-inactivated *P. gingivalis* or *S. sanguinis* bacteria for 48 h by ELISA. Data obtained from ELISA (Appendix A) indicated that the basal expression of S100A15 in control GF and KB cells was very weak that increased significantly following the stimulation of GF and KB cells with either heat-inactivated *P. gingivalis* or *S. sanguinis* bacteria as shown (Appendix A). Taken together, the data of ELISA analysis further confirm the ability of both *P. gingivalis* and *S. sanguinis* bacteria to stimulate the expression of S100A15 protein in oral mucosa-derived cell lines.

### 2.2. Bacterial Pathogens-Induced Expression of S100A15 in Oral Mucosa via Toll-Like Receptor-Dependent Mechanism

LPS mediates its effects via TLR4, whereas LTA mediates its effects via TLR2 [21]. First, we determined the optimal concentration of anti-TLR2 and TLR4 that is able to block microbial pathogens or their purified membrane components LTA/LPS-induced expression of S100A15 as described in the Appendix A. In addition, we investigated whether the concentration of anti-TLR2 or anti-TLR4 influences the cell viability of either GF and KB cells as described in the Appendix A. Finally, we set out to investigate whether S100A15 induction in oral mucosa by gram^−^
*P. gingivalis* or by purified membrane component (LTA) is mediated by TLR4 and if the induction of S100A15 by gram^+^
*S. sanguinis* by purified membrane component (LPS) is mediated by TLR2. GF and KB cells were pre-treated with either TLR2 or TLR4 neutralizing antibodies (anti-TLR2 and anti-TLR4 antibodies) or an isotype IgG control antibody for 30 min prior to the stimulation with heat-inactivated gram^−^ or heat-inactivated gram^+^ bacterial pathogens for 48 h. Cells were harvested for the extraction of total RNAs, and the S100A15 expression was investigated using qRT-PCR. Data obtained from qRT-PCR analysis of S100A15 expression in GF and KB cells following the stimulation with heat-inactivated *P. gingivalis* or heat-inactivated *S. sanguinis* LTA (Figure 2A) demonstrate an inhibition of *P. gingivalis*-induced S100A15 expression in both GF and KB cells in response to pre-treatment with anti-TLR4 antibody, and the inhibition of *S. sanguinis*-induced S100A15 in response to pre-treatment with anti-TLR-2 antibody suggesting the involvement of TLR-4 in the modulation of gram^−^ bacterial pathogens-induced expression of S100A15 and TLR2 in the modulation of gram^+^ bacterial pathogens-induced expression of S100A15. Also, Data obtained from qRT-PCR analysis of S100A15 expression in GF and KB cells following the stimulation with LTA or LPS (Figure 2B) demonstrated the inhibition of LTA-induced S100A15 expression in response to pre-treatment with anti-TLR4 antibody, whereas, the inhibition of LPS-induced S100A15 expression was noted as a consequence for the pre-treatment of both GF and KB cells with anti-TLR-2 antibody, confirming that the induction of S100A15 by gram^−^ microbial pathogens is TLR-4 dependent, whereas, the induction of gram^+^ microbial pathogens is TLR-2-dependent. Taken together, these data demonstrate that the induction of the antimicrobial protein, S100A15, by gram^−^ bacterial pathogens is a TLR4-dependent mechanism, while the induction of the S100A15 protein by gram^+^ bacterial pathogens is a TLR2-dependent mechanism.

### 2.3. Activation of ASK1 and NF-kB Pathways in Oral Mucosa-Derived Cells by Both Gram^−^ and Gram^+^ Bacterial Pathogens

Tumor necrosis factor receptor-associated factor 6 (TRAF6), apoptosis regulating kinase1 (ASK1), and NF-κB are involved in the modulation of TLR2 and TLR4-induced signaling pathways [22,23]. We, therefore, sought to investigate whether the stimulation of TLR2 and/or TLR4 with the bacterial pathogens is associated with the induction of TRAF6 expression and ASK1 activation in the oral mucosa. GF and KB cells were challenged with either heat-inactivated gram^−^
*P. gingivalis* or gram^+^
*S. sanguinis* (2 × 10^4^ cells/mL) for 48 h. Then, total cell lysates were prepared, and a Western blot was used to analyze the expression and/or the phosphorylation of TRAF6, ASK1 (Figure 3). Although the induction of TRAF6 expression and the phosphorylation of ASK1, no alteration was noted at the expression level of ASK1 in GF and KB cells following the stimulation of either gram^−^ (Figure 3A) or gram^+^ (Figure 3B) bacterial pathogens, suggesting an important role for TRAF6 in the modulation TLR2 and TLR4 signal to the downstream signaling pathways including, ASK1 and NF-κB leading to the expression of S100A15 in the oral mucosa.

The role of LPS-induced ASK1 activity in the activation of mitogen-activated protein kinases, including c-Jun N-terminal kinase (JNK) and p38 pathways, has been reported in several studies [24,25]. We, therefore, decided to investigate whether bacterial pathogen-induced ASK1 activity is involved in the activation of MAP kinase signaling pathways, JNK, and/or p38. Both GF and KB cells were stimulated with either heat-inactivated gram^−^ (*P. gingivalis*) or gram^+^ (*S. sanguinis*) bacterial pathogens for 48 h. Then the total cell lysates were prepared. Next, we analyzed the expression and the phosphorylation of JNK and p38 proteins by Western blot using specific antibodies. Although data obtained from Western blot analysis (Figure 3) did not reveal any marked alteration at the expression levels of either JNK or p38 following stimulation of both GF and KB cells with either heat-inactivated *P. gingivalis* or heat-inactivated *S. sanguinis* demonstrated the phosphorylation of JNK and p38 by the gram^−^ (Figure 3A) and gram^+^ (Figure 3B) bacterial pathogens.

To demonstrate whether bacterial pathogen-induced activation of JNK and p38 pathways are associated with the enhancement of the DNA-binding activities of their physiological substrates, namely, AP-1 and/or ATF-2. GF and KB cells were stimulated by heat-inactivated gram^−^
*P. gingivalis* or heat-inactivated gram^+^
*S. sanguinis* for 48 h, and nuclear extracts were prepared from both stimulated and control cells. DNA-binding activities of the transcription factors AP-1 and ATF-2 were observed using EMSA (Figure 3D–G). The DNA-binding activities of the transcription factors AP-1 and ATF-2 were induced in both cell lines by stimulation with heat-inactivated bacterial pathogens, suggesting that activation of AP-1 and ATF-2 involved with gram^−^ and gram^+^ bacterial pathogen-induced activation of JNK and p38.

To address the role of ASK1 in the modulation of bacterial pathogens-induced activation of JNK and/or p38. First, we determined the optimal concentration of the inhibitor of ASK1 (thioredoxin) that is essential to block microbial pathogens or their purified membrane components LTA/LPS-induced activation of ASK1 that may be involved in the modulation of the activation JNK or/and p38 pathways as described in the Appendix A. In addition, we investigated whether the recommended concentration of thioredoxin influences the cell viability of either GF and KB cells as described in the Appendix A. Then, the GF and KB cell lines were pre-treated with ASK1 inhibitor (thioredoxin) 1h prior to the bacterial stimulation. Total cell lysates were prepared and subjected or Western blot analysis to assess the expression and the phosphorylation of both JNK and p38 (Figure 4A,B). Pre-treatment with ASK1 inhibitor resulted in the inhibition of bacterial pathogen-induced phosphorylation of both JNK and p38 with no effect observed on JNK and p38 protein expression, suggesting an important role for ASK1 in the modulation of bacterial pathogen-induced ASK1 phosphorylation is responsible for the activation of JNK and p38 pathways.

We also set out to confirm whether the bacterial pathogen-induced activation of both the JNK and p38 pathways is responsible for the activation of the transcription factors AP-1 and ATF-2, respectively. First, we determined the optimal concentration of the inhibitors of JNK (Sp600125) and p38 (SB-203580) pathways that are essential to blocking microbial pathogens or their purified membrane components LTA/LPS-induced activation of JNK or/and p38 pathways that may be involved in the of S100A15 as described in the Appendix A. In addition, we investigated whether the recommended concentration of the inhibitors of JNK and p38 influences the cell viability of either GF or KB cells as described in the Appendix A. Accordingly, GF and KB cells were pre-treated with the inhibitor of JNK (SP60015) or p38 (SB-203580) 1h prior to bacterial stimulation. Then nuclear extracts were prepared and subjected to EMSA analysis. Inhibition of bacterial pathogen-induced AP-1 activation AP-1 was observed in response to pre-treatment of both GF and KB cells with the JNK inhibitor (Figure 4E,F), and ATF-2 inhibition was observed following p38 inhibition (Figure 4G,H), suggesting that the induced activation of JNK and P38 is responsible for the activation of AP-1 and ATF-2, respectively.

Next, we decided to address the role of the NF-κB pathway in the modulation of gram^−^ and gram^+^ bacterial pathogens-induced effects in both GF and KB cells. Both GF and KB cell lines were stimulated with either heat-inactivated *P. gingivalis* or heat-inactivated *S. sanguinis* for 48 h. The total cell lysate was collected for Western blot and kinase assay, as well as nuclear and cytoplasmic protein extraction for EMSA and Western blot analysis. Data obtained from IκBα kinase assays show that inhibition of nuclear factor kappa-B kinase subunit alpha (IKKα) impacts IκBα’s ability to phosphorylate in vitro the NF-κB inhibitor in both GF and KB cells following stimulation with either heat-inactivated *P. gingivalis* (Figure 5A) or heat-inactivated *S. sanguinis* (Figure 5B), suggesting an essential role for the NF-κB pathway in the modulation of both gram^−^ and gram^+^ bacterial pathogens-induced effects in the oral mucosa. In addition, degradation levels analysis of IκBα protein by Western blot using the cytoplasmic protein of control and stimulated cells demonstrated the reduction of IκBα protein levels in both GF and KB cells in response to the stimulation with either heat-inactivated *P. gingivalis* (Figure 5A) or heat-inactivated *S. sanguinis* (Figure 5B) providing evidence for the activation of NF-κB by bacterial pathogens in oral mucosa-derived cell lines. Likewise, the analysis of the nuclear extracts demonstrated the enhancement of the DNA-binding activity of NF-κB in both GF and KB cells following stimulation with either *P. gingivalis* (Figure 5D) or *S. sanguinis* (Figure 5E), suggesting an important role for the activated NF-κB pathway in the regulation of S100A15.

### 2.4. ASK1-JNK, ASK1-p38, and NF-κB Pathways Are Involved in the Modulation of Bacterial Pathogens-Induced Expression of S100A15 in Oral Mucosa

To show whether bacterial pathogens-induced activation of ASK1-JNK, ASK1-p38, and NF-κB pathways are involved in the regulation of the antimicrobial protein S100A15, GF, and KB, the cells were pre-treated with the inhibitors of ASK1 (thioredoxin), JNK (SP600125) or p38 (SB-203580) 1 h prior to the stimulation with heat-inactivated *P. ginigvalis* or heat-inactivated *S. sanguinis* for 48 h. Then, treated and control cells were harvested, and the total RNA was extracted and subsequently subjected to the analysis of S100A15 expression using qRT-PCR. Inhibition of heat-inactivated bacterial pathogens (*P. ginigvalis* or *S. Sanguinis*)-induced expression of S100A15 in both GF and KB cells by the pre-treatment with ASK1 inhibitor (Figure 6A), providing evidence for the involvement of ASK1 in the modulation of both *P. ginigvalis* and *S. Sanguinis*-induced S100A15 expression in the oral mucosa. To investigate whether bacterial pathogens-induced activation of JNK, p38, and NF-κB pathways are essential for the modulation of the induced expression of S100A15. GF and KB cells were pre-treated with the inhibitors of the JNK, p38, and NF-κB pathways individually or in combination 1 h prior to the stimulation with either heat-inactivated *P. gingivalis* or heat-inactivated *S. sanguinis* for 48 h (Figure 6A,C). A partial inhibition of bacterial pathogens-induced mRNA of S100A15 by the specific inhibitors of either JNK (SP600125), p38 (SB-203580), or NF-kB (Bay11-7082), whereas the combination of the three inhibitors were found to abrogate heat-inactivated *P. gingivalis*-induced S100A15 expression (Figure 6B) as well as heat-inactivated *S. sanguinis*-induced expression of S100A15 (Figure 6C), suggesting the involvement of JNK, p38 and NF-kB pathways in the regulation of bacterial pathogens-induced expression of S100A15 in the oral mucosa.

### 2.5. Verification of the Clinical Validity of Both Gram^−^ and Gram^+^ Bacterial Pathogens-Induced Expression of S100A15 Protein

We next examined if the S100A15 protein is expressed in the gingival tissue of patients with inflammatory gingiva to verify the clinical validity, as well as to investigate whether the induced expression of S100A15 is attributable to the colonization of gram^−^ and/or gram^+^ bacterial *pathogens* in the oral cavity. The expression of the S100A15 protein in healthy individuals (n = 3) and patients with inflammatory gingiva (n = 3) was observed via immune histochemistry using a specific antibody. Concurrently, a specific DNA marker for gram^−^
*P*. *gingivalis* and for gram^+^
*S. sanguinis* was employed for the detection of both gram^−^ and gram^+^ bacterial pathogens in patients’ saliva using conventional PCR. The expression of S100A15 was undetectable in the gingiva of the control individuals (Figure 7A), whereas its expression was significantly elevated in patients with inflammatory gingiva (Figure 7B), suggesting that S100A15 expression may be the consequences of gingiva inflammation by oral microbial pathogens. To investigate whether the inflammation of the gingiva and, subsequently, the elevated expression of S100A15 is attributed to the colonization of bacterial pathogens in the oral cavity, we set out to identify the possible bacterial pathogens in the saliva of the patients with inflammatory gingiva (n = 3) by the amplification of the 480 bp fragment of the 16S rRNA gene as a marker for *P. gingivalis* (Figure 7D) and the amplification of the 473 bp fragment of the UDP-N-acetylglucosamine-like protein gene as a marker for *S. sanguinis* (Figure 7E) by PCR. PCR analysis shows the colonization of gram^−^
*P. gingivalis* (Figure 7D) and gram^+^
*S. sanguinis* (Figure 7E) in saliva collected from patients with inflammatory gingiva, suggesting that the colonization of both gram^−^ and gram^+^ bacterial pathogens are the main cause for the induction of S100A15 protein in the oral mucosa. Taken together, the expression of S100A15 protein in oral mucosa tissues and the identification of the bacterial pathogens, namely, *P. gingivalis* and *S. sanguinis,* provide evidence for the involvement of both gram^−^ and gram^+^ bacterial pathogens in the induction of S100A15 in oral gingiva.

## 3. Discussion

In this study, we demonstrate for the first time that induction of S100A15 in the oral mucosa is attributable to the colonization of both gram^−^ and gram^+^ bacteria in the oral cavity and provide insight into the mechanisms which are implicated in the transcriptional regulation of S100A15 protein in the oral mucosa.

Although many AMPs, including S100A15, have been reported in several studies for their contributing role in host innate immunity, particularly in herpes simplex virus-and Epstein-Barr virus-infected oral mucosa [26,27], the mechanisms of S100A15 in the innate immunity of oral mucosa, so far, is not discussed in detail.

In the present study, we demonstrate for the first time that the induction of S100A15 expression in oral mucosa by either gram^−^ (*P. gingivalis*) or gram^+^ (*S. Sanguinis*) bacteria is mediated by TLR4 and TLR2, respectively. The stimulation of both TLR4 and TLR2 signals by either gram^−^ or gram^+^ bacteria is mediated via TRAF6, leading to the activation of NF-κB and ASK1-JNK-AP-1 and ASK1-p38-ATF-2 pathways to induce the expression of S100A15 in the oral mucosa. These findings indicate that the production of antimicrobial proteins (AMPs) such as S100A15 by the epithelial cells in oral mucosa belongs to the host’s innate immunity, which overall may aid the oral mucosa in its protection against oral microbial pathogens.

In mammalian innate immunity, the recognition and processing of microbial pathogens are mediated by Toll-like receptors (TLRs)-dependent mechanism. This mechanism is initiated in the form of an extracellular signal leading to the activation of intracellular signaling pathways, including NF-κB and ASK1- dependent mitogen-activated protein (MAP) kinases, namely, JNK and p38 [28,29]. The role of ASK1 in the regulation of MAP kinase kinase 4 (MKK4)-JNK and MAP kinase kinase 7 (MKK7)-JNK, MAP kinase kinase 3 (MKK3)-p38 and MAP kinase kinase 6 (MKK6)-p38 pathways have been widely documented in several studies [30,31,32].

The findings of this study indicate that induction of S100A15 protein by gram^−^ or gram^+^ bacterial pathogens, or even by their purified membrane components, is a toll-like receptor-dependent mechanism. The induction of S100A15 expression by either gram^−^ bacterial pathogens (e.g., *P. gingivalis*) or by their purified membrane components (LPS) is mediated by TLR4, while its induction by either gram^+^ bacterial pathogens (e.g., *S. sanguinis*) or their purified membrane components (LTA) is mediated by TLR2. Although the stimulation of oral mucosa by gram^−^ or gram^+^ bacterial pathogens is mediated by two independent and quite different Toll-like receptors, the transcriptional regulation of S100A15 in oral mucosa is mediated by the same molecular mechanism. This mechanism is attributed to the ability of the stimulated TLR4 and TLR2 to mediate their signals via TRAF6, leading to the activation of the NF-κB pathway together with ASK1-dependent kinases in the form of ASK1-JNK and ASK1-p38 pathways. The activation of the three pathways, namely NF-κB, ASK1-JNK-AP-1, and ASK1-p38-ATF-2, leads to the formation of a transcriptional complex that consists of the three transcription factors AP-1, ATF-2, and NF-κB. The binding of the transcriptional complex to the corresponding DNA-binding sites of the promoter triggers the transcription of the gene encoding for antimicrobial protein S100A15.

In this study, the inhibitory experiments confirmed that bacterial pathogens-induced TLR2 and TLR4 signals to TRAF6 leading to induction of S100A15 protein is mediated by NF-κB and ASK1 pathways. Moreover, the abrogation of the induced expression of S100A15 by the inhibition of the three pathways, namely, ASK1-JNK-AP-1, ASK1-p38-ATF-2 and NF-κB pathways, indicates that the activation of the three pathways is essential for transcriptional regulation of S100A15 gene.

Taken together, in this study, we demonstrated that the induction of S100A15 in oral mucosa by gram^−^ bacterial pathogens is mediated by TLR4, while its induction by gram^−^ bacterial pathogens is mediated by TLR2. In addition, we provided insight into the mechanisms of the induction and the transcription regulation of S100A15 by oral bacterial pathogens in oral mucosa is mediated by three pathways, namely, ASK1-JNK-AP1, ASK1-p38-ATF-2, and NF-κB pathways. Thus, based on the outcome of the present study, we have proposed a diagram (Figure 8) that outlines, in detail, the mechanisms of gram^−^ and gram^+^ bacterial pathogens-induced expression of S100A15 in the oral mucosa.

## 4. Materials and Methods

### 4.1. Stimulation of Oral Mucosa-Derived Cells Lines by LPS, LTA, and Bacterial Pathogens

Both human Gingival Fibroblast, GF (ATCC^®^ PCS-201-018™) and Mouth Epidermal Carcinoma, KB (ATCC^®^, CCL-17) cell lines were grown as monolayers culture under standard conditions in DMEM medium containing 10% FCS. For in vitro stimulation, *P. gingivalis* (ATCC^®^ 33277) and *S. sanguinis* (ATCC^®^ 10556) bacterial cultures were allowed to grow in Luria-Bertani (LB) medium at 37 °C, and the optical density (OD) was measured at 578 nm (OD578). Bacteria were then harvested and washed twice in PBS, diluted to working solutions, and heat-inactivated at 68 °C for 20 min. For indicated experiments, bacteria were further centrifuged at 3000× *g* for 20 min. To separate the membrane components of the bacterial pathogens. Then, the heat-killed bacterial compounds were used to stimulate GF and KB cells corresponding to approximately 2 × 10^4^ CFU/mL for *S. sanguinis* and *P. gingivalis* as well, per well. Whereas LPS concentrations of 100 ng/mL (Sigma-Aldrich; St. Louis, MO, USA) and LTA concentrations of 50 ng/mL were used for the stimulation of GF and KB cells.

### 4.2. RNA Preparation, Semiquantitative and Real-Time PCR

Total RNAs were extracted from both GF and KB cells using the RNAeasy Minikit (Qiagen) according to the manufacturer’s protocol. S100A15 expression was analyzed by semi-quantitative RT-PCR using the following primer pairs: S (5′-ACG TCA CTC CTG TCT CTC TTT GCT-3′) and AS (5′-TCA TGA ATC AAC CCA TTT CCT GGG-3′) as well as β-actin primer pairs: S (5′-AGA GAT GGC CAC GGC TGC TT-3′) and AS (5′-ATT TGC GGT GGA CGA TGG AG-3′. RT-PCR was performed in duplicate. The PCR conditions were 35 cycles for S100A15 and -actin at 94 °C for 1 min, and (65 °C for S100A15 and 61 °C for β-actin) 1 min, 72 °C for 1 min followed by an additional extension step at 72 °C for 10 min. A negative control was set up for each reaction. Amplicons were subjected to electrophoresis in a 2% agarose gel containing ethidium bromide, and the separated S100A15 (45 bp) and Actin (600 bp) were photographed and scanned. The densitometric quantification of S100A15 mRNA over actin mRNA transcript was presented as the relative ratio of S100A15 band and actin.

Analysis of S100A15 by qRT-PCR was performed as previously described [33]. The first-strand cDNA synthesis was performed using a High-Capacity RNA-to-cDNA™ Kit (Applied Biosystems™, Waltham, MA, USA; #4387406). cDNA synthesis was primed with a mixture of oligo dT and random oligomers (allowing the cDNA synthesis of 18S rRNA) using 1 μg of total RNA. RNA was then incubated at 65 °C for 5 min and chilled on ice before the addition of the remaining components in the following reaction mixture (20 μL): 50 mM Tris–HCl pH 8.3, 75 mM KCl, 3 mM MgCl2, 10 mM dNTP, 40 U RNAse inhibitor (Invitrogen, Carlsbad, CA, USA), 200 U Superscript II (Invitrogen, Carlsbad, CA, USA) and incubated for 1 h at 42 °C. The reaction was heat-inactivated at 70 °C for 10 min followed by incubation at 4 °C until use.

Analysis of S100A15 by qRT-PCR was performed as previously described [33], using SYBR Green Master Mix kit from Applied Bioscience (ABI, Foster City, CA, USA); probes for the amplification of S100A15 (forward primer: 5′-AGC AAC ACT CAA GCT GAG AGG-3′ and reverse primer: 5′-TCC ATG GCT CTG CTT GTG GTA-3′), 18S (forward primer: 5′-CGG AGG TTC GAA GAC GAT CA-3′ reverse primer: (5′-CAT CGT TTA TGG TCG GAA CTA CG-3′). The total reaction volume was 20 μL containing 6 μL cDNA template, 10 μL of 2 × SYBR Green master mix (# 4385616; Applied Biosystems, Waltham, MA, USA), 0.5 μL (5 μM) primer Forward, and 0.5 μL (5 μM) primer Reverse of the S100A15, and 0.5 μL (5 μM) primer Forward and 0.5 μL (5 μM) primer Reverse of the 18S rRNA and 2 μL H2O. RT-qPCR was performed under the following conditions: an initial denaturation step for 20 s at 95 °C, followed by 40 cycles of amplification with 5 s of denaturation at 95 °C, 30 s of annealing, and extension at 55 °C. A non-template control (NTC) was also included in each run for each gene. 18S rRNA gene was used as a housekeeping gene to normalize RT-qPCR data. The expression was evaluated using an SYBR Green protocol, according to the manufacturer’s instructions. The Analysis of S100A15 gene expression was performed in triplicate in an ABI Prism 7000 Sequence Detection System. Fold induction relative to the vehicle-treated control was calculated using the 2(-Ct) method, where Ct is Ct (stimulant)-Ct (control), Ct is Ct (target gene)-Ct (housekeeping gene), and CT is the cycle at which an arbitrary detection threshold is crossed. Data are presented as means ± SD for S100A15.

### 4.3. Immunoblot

Immunoblot analysis was performed according to the standard procedures [34]. The following antibodies were used at the indicated dilution: Anti-ASK1 (SC-7931) 1:500; antibody; anti-JNK (Sc-474), 1:2000; anti-p38 (Sc-535), 1:2000; anti-Actin (SC-1615), 1:5000 (Santa Cruz Biotechnology, Inc., Dallas, TX, USA), Anti-TRAF6 [EP591Y] 1:500; antibody (ab33915, Abcam, Cambridge, UK), anti-S100A15 (Cat: 50226-RP01) 1:500; antibody (Sino Biological, Bejing, China).

### 4.4. Preparation of Nuclear Extracts

Both GF and KB cells were plated into a 10 cm dish (Nunc) and cultured in a medium for 48 h prior to stimulation with oral gram^−^/gram^+^ microbial pathogens or their purified membrane components. Cells were harvested at indicated time periods, and the nuclear extracts were prepared as described [35]. Briefly, cells were washed with ice-cold PBS buffer and harvested by the addition of 500 µL of buffer A (20 mM HEPES, pH 7.9; 10 mM NaCl; 0.2 mM EDTA; and 2 mM DTT) containing protease inhibitor and incubated on ice for 10 min. The supernatant was discarded after centrifugation at 14,000× *g* rpm for 3 min. The pellet was resuspended in 50 μL of buffer C (20 mM HEPES, pH 7.9; 420 mM NaCl; 0.2 mM EDTA; 2 mM DTT; 1 mM Na_3_OV_4_ and 25% glycerol) containing protease inhibitor and incubated for 20 min at 4 °C and then centrifuged at 14,000× *g* rpm for 3 min. The supernatant was collected and stored at −80 °C until use.

### 4.5. Protein Extraction and Enzyme-Linked Immunosorbent Assay (ELISA)

The total protein was extracted from GF and KB cells cultured under normal conditions in 150 mm Petri dishes for 24 h before the stimulation with 2 × 10^4^ cells of either heat-inactivated *P. gingivalis* or *S. sanguinis* bacteria. Forty-eight hours later, treated and control cells were subjected to the extraction of the total protein using 1 mL of lysis buffer (150 mM NaCl; 50 mM Tris-HCl, pH 7.5; 1% Triton X-100; and 0.25% sodium deoxycholate). Then, the detection of S100A15 protein in both control and stimulated cells was performed by Sandwich enzyme immunoassay using an S100A15 ELISA kit (Cat#EKN48277, BioMatik corporation, Wilmington, DE, USA) according to the manufacturer’s protocol. Briefly, the microtiter plate provided in this kit has been pre-coated with an antibody specific to S100A15. 100 μL (~5 µg) of total protein extraction of control and stimulated cells were then added to the microtiter plate wells with a biotin-conjugated antibody specific to S100A15 protein. Next, Avidin conjugated to Horseradish Peroxidase (HRP) was added to each microplate well and incubated for 1 h at RT with gentle continual shaking (~500 rpm). After the aspiration of the content, the wells were washed with 300 μL of washing buffer 5 times. After washing had been done, 100 μL of the detection antibody solution was added into each well and allowed to incubate for two hours at RT with gentle continual shaking (~500 rpm). At the end of the incubation, the contents were aspirated, and the wells were washed five times with >300 μL of Wash buffer per well. After the washing had been done, a 100 μL of 3,3,5,5-Tetramethylbenzidine (TMB) solution was added to each well. After the incubation of the plate for 30 min at RT. Finally, 50 μL of stop solution was added. Optical densities of kit standards and test samples were read at 450 nm using a microplate reader. All samples had six repeats in each experiment. The concentrations of test samples were calculated according to the relation between absorbance values and concentrations of test standards.

### 4.6. Electrophoretic Mobility Shift Assay (EMSA)

EMSAs were performed as described [35]. Double-stranded synthetic oligonucleotides carrying binding sites for ATF-2, AP-1, NF-κB (Santa Cruz, Lüdinghausen, Germany) were end-labeled with [γ-^32^P] dATP (Hartmann Analytika, Munich, Germany) in the presence of T4 polynucleotide kinase (Genecraft, Lüdinghausen, Germany). For binding, 4 µg nuclear extract was bound to a 0.2 ng probe in a total volume of 30 μL for 30 min at room temperature in binding buffer (10 mM Tris, pH 7.5; 50 mM NaCl, 1 mM EDTA; 1 mM MgCl_2_; 0.5 mM DTT and 4% glycerol). The specificity of the binding was analyzed by competition with an unlabeled oligonucleotide. The competition assay was performed in the same manner, except that unlabeled probes containing ATF-2, AP-1, and NF-κB sequences were incubated with nuclear extracts for 20 min at room temperature before the addition of labeled probes. Electrophoresis was performed for 3 h at 100 V in 0.5 X Tris-borate-EDTA running buffer at room temperature. The dried gel was visualized by exposure to high-performance autoradiography film.

### 4.7. In Vitro Kinase Assays

In vitro kinase assay was performed as described previously [35]. Briefly, both GF and KB cells were allowed to grow for 24 h before the stimulation with LPS or oral microbial pathogens for 48 h. The cells were harvested at the indicated time points, and the total cell lysates were prepared using 500 μL of buffer L (20 mM HEPES [pH 7.9], 10mM EGTA, 40 mM ß-glycerophosphate, 25 mM MgCL_2_, 2 mM Na_3_VO_4_, 1 mM DTT, 1% NP-40, 5 μg apoprotein, 1 mM Leupeptin, 1 μg/mL, pepstatin, and 1 mM benzamidine). Insoluble material was removed by centrifugation, and the cell lysate was incubated with anti-IKKα (C-6) antibody (SC-166231) Santa Cruz, Dallas, TX, USA for 1 h at 4 °C. The immune complexes were bound to A-Sepharose (5 mg/mL in lysis buffer) by rotating overnight at 4 °C. After centrifugation, the A-Sepharose beads were washed three times with kinase reaction buffer (80 mM HEPES [pH 7.9], 80 mM MgCL_2_, 0.1 mM ATP; 2 mM Na_3_OV_4_ and 20 mM NaF). Kinase activity was determined by incubation with 2 μg of GST-IκBa (1-317) protein (SC-4094) Santa Cruz protein as a substrate for IKKα and 10 μCi of [γ-^32^P] dATP (Hartmann Analytika) in 15 μL of kinase reaction buffer and incubated for 30 min at 37 °C. Reactions were terminated by the addition of 15 μL of sample buffer and analyzed by SDS-polyacrylamide gel electrophoresis. The gel was dried and autoradiographed.

### 4.8. Collection and Processing of Saliva Samples for PCR

The collection of saliva samples (100 μL each) was taken from the base of the mouth of patients with gum inflammation (n = 10) and approved according to the institutional review board protocol (IRB) of the University Hospital of Aachen. All experimental procedures were approved by the Ethics Committee of the University Hospital of Aachen, and written consent forms were obtained from the participants. A 30 μL aliquot of each saliva specimen was boiled for 10 min and then centrifuged at 10,000× *g* for 5 min, and 5 μL of supernatant was used as a template for PCR.

### 4.9. Amplification of PCR

The amplification of PCR to confirm the presence of the oral microbial pathogens *P. gingivalis* and *S. sanguinis* in the collected saliva was performed as follows: two *P. gingivalis*-specific primer pairs were used to amplify a 480 bp fragment of the 16S rRNA gene: S primer (5′-TTG CCC GAA AAA TTA AGG AGA-3′) and AS primer (5′-ACT GCT TCT CTC TTA TTC GAA-3′). *S. sanguinis*-specific primer pairs, S primer (5′-AAG CCA TTT TGC CTA GAT TGA-3′) and AS primer (5′-CAT ACC GAT TCC TTA CTC TAA-3′) were used to amplify a 473 bp fragment of the UDP-N-acetylglucosamine-like protein gene. The amplification reactions were performed in a total volume of 50 μL consisting of 0.2 mM each deoxynucleoside triphosphate (dATP, dTTP, dCTP, and dGTP), 10× *Taq* buffer (Genecraft, Lüdinghausen, Germany), 1 μM each primer, 2.5 U of Taq polymerase (Genecraft), and 0.5 to 6 μL of the template. PCR amplification was performed in a thermocycler (BioRad). Cycling parameters were as follows: an initial denaturation at 95 °C for 1 min; 35 cycles consisting of 95 °C for 30 s, 49 °C for 1min, and 72 °C for 1 min, and a final extension at 72 °C for 10 min. *P. gingivalis* (ATCC 33277) and *S. sanguinis* (ATCC 10556) cells (50 cells per PCR) were used as a positive control, and 5 μL of water constituted the negative control. Positive and negative controls were included in each PCR set and in all sample processing. The amplification products and the positive controls were subjected to electrophoresis in a 2% agarose gel containing ethidium bromide (0.5 μg/mL) and photographed under UV illumination using the Polaroid MP4 system. A 1-kb DNA ladder (Genecraft, Lüdinghausen, Germany) was used as a molecular size standard.

### 4.10. Immune Histochemistry

Gingival tissue biopsy specimens of either healthy individuals or patients with inflammatory gingiva were harvested from extracted teeth from either the buccal or lingual site and according to the approved institutional review board protocol (IRB) of the University Hospital of Aachen. The experimental procedures were approved by the Ethics Committee of the University Hospital of Aachen, and written consent forms were obtained from the participants. The biopsies were fixed in 4% buffered formalin and were transported to the laboratory, where the specimens were embedded in paraffin wax and sectioned. The expression of S100A15 in gingival tissues was detected by immunohistochemistry using an S100A15 antibody according to standard methods [36]. Antigen retrieval was carried out by immersing the slides in a whole buffer and incubated in an enzyme retriever (Biogenex, Fremont, CA, USA) at 95 °C for 10 min. Endogenous peroxidase activity was blocked with 3% hydrogen peroxide in phosphate-buffered saline (PBS) containing Tween [PBS, 50 mm sodium phosphate pH 7.6, 200 mm sodium chloride (NaCl), and 0.1% Tween 20] for 15 min. and rinsed, and nonspecific binding of immunoglobulin G (IgG) was blocked with 5% normal swine serum (NSS, lot 107, Dako A/S, Glostrup, Denmark) in PBS plus Tween. The primary for S100A15 was diluted in 5% NSS in PBS plus Tween (1 μg/mL) and incubated at 20 °C for 1 h. The antibody was diluted in 5% NSS in PBS plus Tween. The sections were rinsed again and overlaid with the secondary antibody for 45 min. Immunoperoxidase labeling was performed using the avidin-biotin complex (ABC). The sections were then incubated with chromogen (3.3′-diaminobenzine tetrahydrochloride 0.05%, Dako A/S, Glostrup, Denmark) and 0.001% hydrogen peroxide in PBS. The process was stopped with cold tap water, and the stained slides were screened, and the areas for examination were determined by a non-biased observer using a light microscope (Leica DMRB) equipped with a digital video camera. Each section was analyzed for S100A15-positive stained cells.

### 4.11. Statistical Analysis

Values are expressed as the mean ± standard deviation. Statistical analysis was performed using SPSS (version 22) software for Windows. All analyses used two-sided hypothesis tests. The differences between experimental groups were determined using the *t*-test and one-way analysis of variance (ANOVA).

## 5. Conclusions

Taken together, in this study, we demonstrated that the induction of S100A15 in oral mucosa by gram^−^ bacterial pathogens is mediated by TLR4, while its induction by gram^+^ bacterial pathogens is mediated by TLR2. In addition, we provided insight into the mechanisms of the induction and the transcription regulation of S100A15 by oral bacterial pathogens in oral mucosa is mediated by three pathways, namely, ASK1-JNK-AP1, ASK1-p38-ATF-2, and NF-κB pathways. We have proposed a diagram (Figure 8) that outlines, in detail, the mechanisms of gram^−^ and gram^+^ bacterial pathogens-induced expression of S100A15 in the oral mucosa.

## Figures and Tables

**Figure 1 ijms-24-05348-f001:**
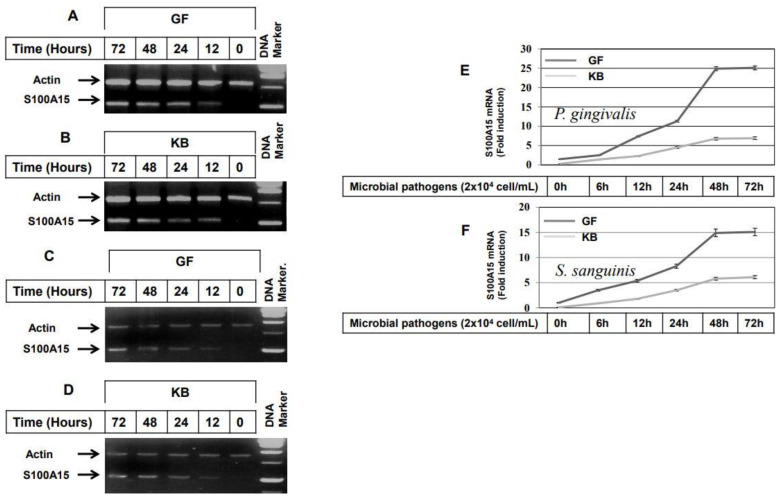
Induction of S100A15 expression in oral mucosa by both gram^−^ and gram^+^ oral bacterial pathogens. (**A**–**D**) RT-PCR expression of S100A15 by the stimulation of GF (**A**) and KB (**B**) cells with lipopolysaccharides (LPS) at a concentration of 100 ng/mL for regulated time intervals up to 72 h; expression of S100A15 by the stimulation of GF (**C**) and KB (**D**) cells with lipoteichoic acid (LTA) at a concentration of 50 ng/mL for regulated time intervals up to 72 h. Total RNAs were extracted from treated cells at different time intervals and subjected to RT-PCR analysis as described under material and methods. The RT-PCR products were analyzed on 2% agarose stained with ethidium bromide. The leader DNA 1.0 kbp was used as a DNA marker. Actin was used as an internal control. Data are representative of three independent experiments performed separately. (**E**,**F**) qRT-PCR expression of S100A15 by the stimulation of GF and KB cells with either heat-inactivated *P. gingivalis* (**E**) or heat-inactivated *S. sanguinis* (**F**) microbial pathogens (MPs) for regulated time intervals up to 72 h. Total RNA was extracted from treated cells at different time points and subjected to qRT-PCR analysis as described under material and methods. The represented data are the mean ± SD of three independent experiments performed in duplicate.

**Figure 2 ijms-24-05348-f002:**
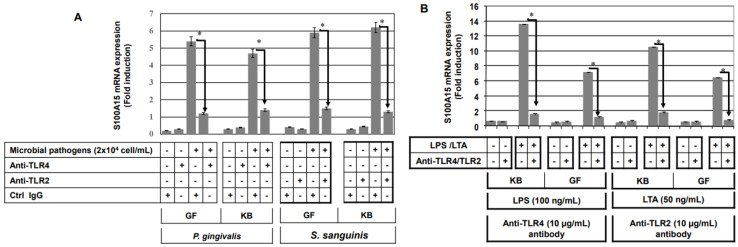
Inhibition of induced expression of S100A15 by neutralization of toll-like receptors. (**A**) GF and KB cells were pre-treated with anti-TLR2, anti-TLR4, or control IgG 1 h prior to the stimulation with either heat-inactivated *P. gingivalis* or heat-inactivated *S. sanguinis* for 48 h. Total RNA was extracted from treated and control cells and subsequently subjected to qRT-PCR analysis to assess the relative transcription levels of S100A15. (**B**) GF and KB cells were pre-treated with anti-TLR2, anti-TLR4, or control IgG 1 h prior to stimulation with either LPS (100 ng/mL) or LTA (50 ng/mL) for 48 h. Total RNAs were extracted from treated and control cells and subsequently subjected to qRT-PCR analysis to assess the relative transcription levels of S100A15. Densitometric quantification of S100A15 mRNA over actin mRNA transcript. The data were normalized to the level of treated and untreated control cells in each sample. Each bar represents the mean ± SD of three independent experiments performed in duplicate, * *p* < 0.05.

**Figure 3 ijms-24-05348-f003:**
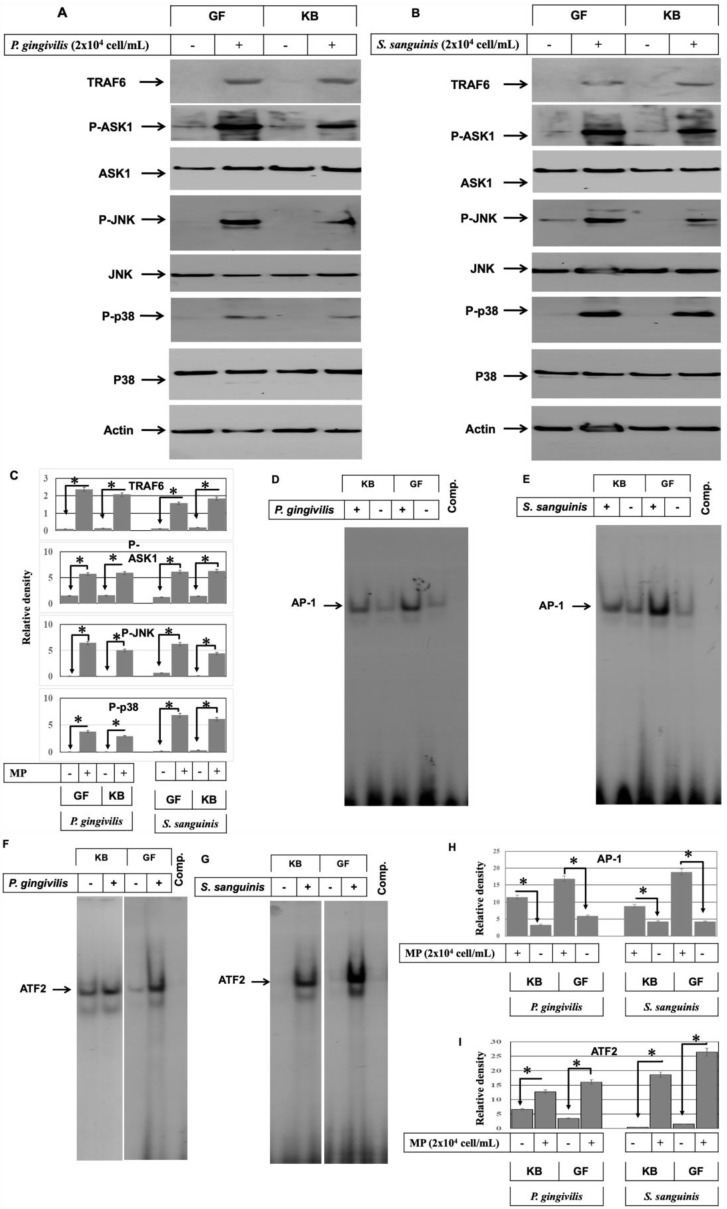
Induction of ASK1, JNK, and p38 pathways in oral mucosa by both gram^−^ and gram^+^ bacterial pathogens. Both GF and KB cells were stimulated with either heat-inactivated *P. gingivalis* (2 × 10^4^ cells/mL) or *S. sanguinis* (2 × 10^4^ cells/mL) microbial pathogens (MPs) for 48 h (**A**,**B**). Total protein lysate and nuclear extract were prepared to perform kinase assays, Western blot, and EMSA. Western blot demonstrates the induction of TRAF6 expression and phosphorylation of ASK1 activity in both GF and KB cells by stimulation with either heat-inactivated *P. gingivalis* (**A**) or heat-inactivated *S. sanguinis* (**B**) when compared with control cells. Also, Western blot analysis of the total lysates of treated and control cells using anti-phospho-ASK1, phospho- JNK, phospho-p38, TRAF6, ASK1, JNK, or p38 antibodies demonstrate the induction of JNK and p38 phosphorylation without any marked alteration to the expression level of ASK1, JNK, or p38 by stimulation with either heat-inactivated *P. gingivalis* (**A**) or heat-inactivated *S. sanguinis*(**B**), when compared with control cells. Actin was used as an internal control for loading and transfer. (**C**) Analysis of band intensity on films is presented as the relative ratio of TRAF6 expression to the expression of actin, the relative ratio of phospho-ASK1, Phospho-JNK, and Phospho-p38 to the expression of ASK1, JNK, and p38 expression. The data were normalized to the level of Actin expression in treated and untreated control cells in each sample. Each bar represents the mean ± SD of three independent experiments performed separately, * *p* < 0.05. (**D**,**E**) The DNA-binding activity of the transcription factors AP-1 and ATF-2 were investigated by EMSA using nuclear extracts of treated and control cells. The stimulation of both GF and KB cells with either heat-inactivated *P. gingivalis* (**D**) or heat-inactivated *S. sanguinis* (**E**) induced the DNA-binding activity of the transcription factor AP-1 when compared with control cells. Also, the stimulation of both GF and KB cells with either heat-inactivated *P. gingivalis* (**F**) or heat-inactivated *S. sanguinis* (**G**) induced the DNA-binding of the transcription factor ATF-2 when compared with control cells. (**H**,**I**) Analysis of band intensity of the transcription factors AP-1 and ATF-2. The data were normalized to the level of the binding activity of the transcription factors in control cells in each sample. Each bar represents the mean ± SD of three independent experiments performed separately, * *p* < 0.05.

**Figure 4 ijms-24-05348-f004:**
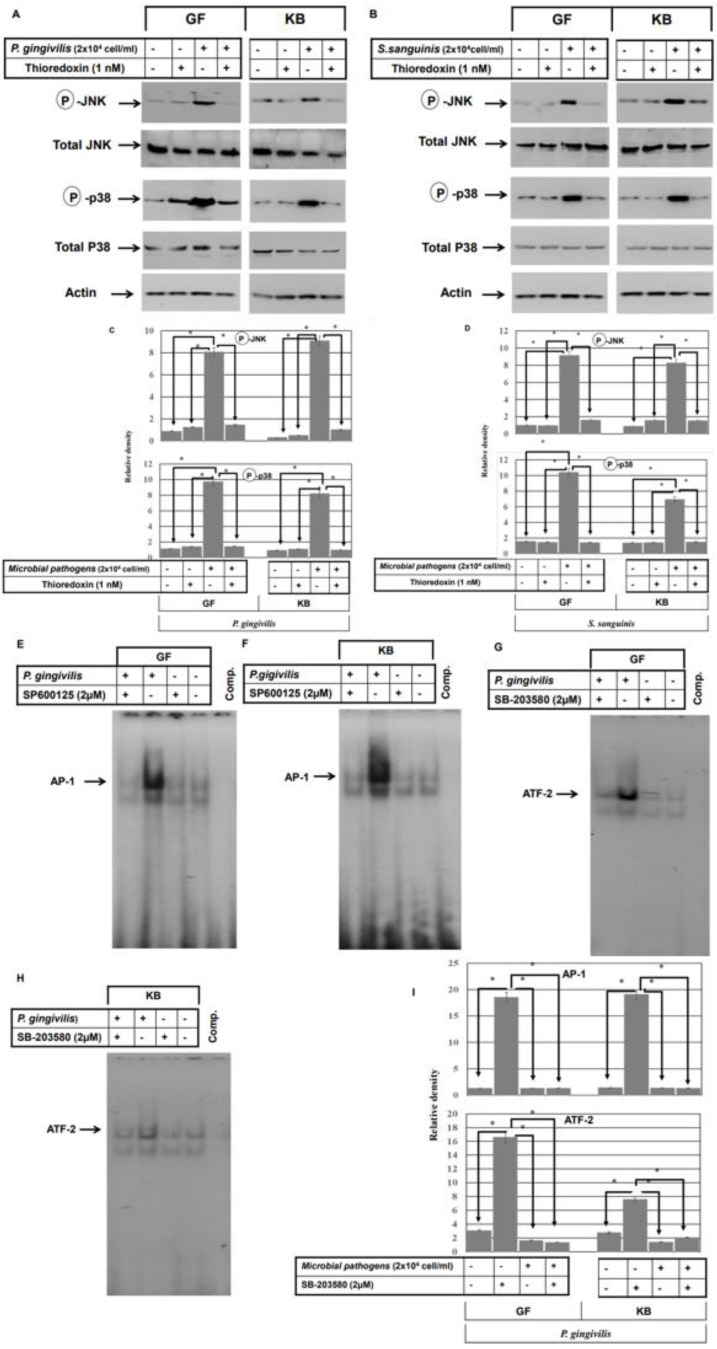
ASK1 mediates the activation of both the JNK and p38 pathways in oral mucosa by stimulation with bacterial pathogens. (**A**–**F**) Both GF and KB cells were pre-treated with the inhibitor of ASK1 (thioredoxin) 1 h prior to stimulation with either heat-inactivated *P. gingivalis* (2 × 10^4^ cells/mL) or heat-activated *S. sanguinis* (2 × 10^4^ cells/mL) microbial pathogens (MPs) for 48 h. Total protein lysate and nuclear extract were then prepared for Western blot analysis and EMSA assay. (**A**,**B**) Western blot analysis of the total lysate of treated and control cells using anti-phospho-JNK, phospho-p38, JNK, or p38 antibodies demonstrate the inhibition of inactivated *P. gingivalis* (**A**) and *S. sanguinis* (**B**)-induced phosphorylation of both JNK and p38 proteins by the inhibitor of ASK1 in both GF and KB cells when compared with control cells. JNK, p38, and actin were used as internal controls for loading and transfer. (**C**,**D**) Analysis of band intensity on films is presented as the relative ratio of Phospho-JNK and Phospho-p38 to the expression of JNK and p38 expression. Bars represent the mean ± SD from three independent experiments performed separately, * *p* < 0.05. (**E**,**F**) Pre-treatment of both GF and KB cells with ASK1 inhibitor suppressed the induced DNA-binding activity of the transcription factors AP-1 (**C**,**D**) and ATF-2 (**E**,**H**) by the stimulation of both GF and KB cells with either heat-inactivated *P. gingivalis* (**E**,**G**) or heat-inactivated *S. sanguinis* (**F**,**H**) bacterial pathogens. (**I**) Analysis of band intensity of the transcription factors AP-1 and ATF-2. Bars represent mean ± SD from three independent experiments performed separately, * *p* < 0.05.

**Figure 5 ijms-24-05348-f005:**
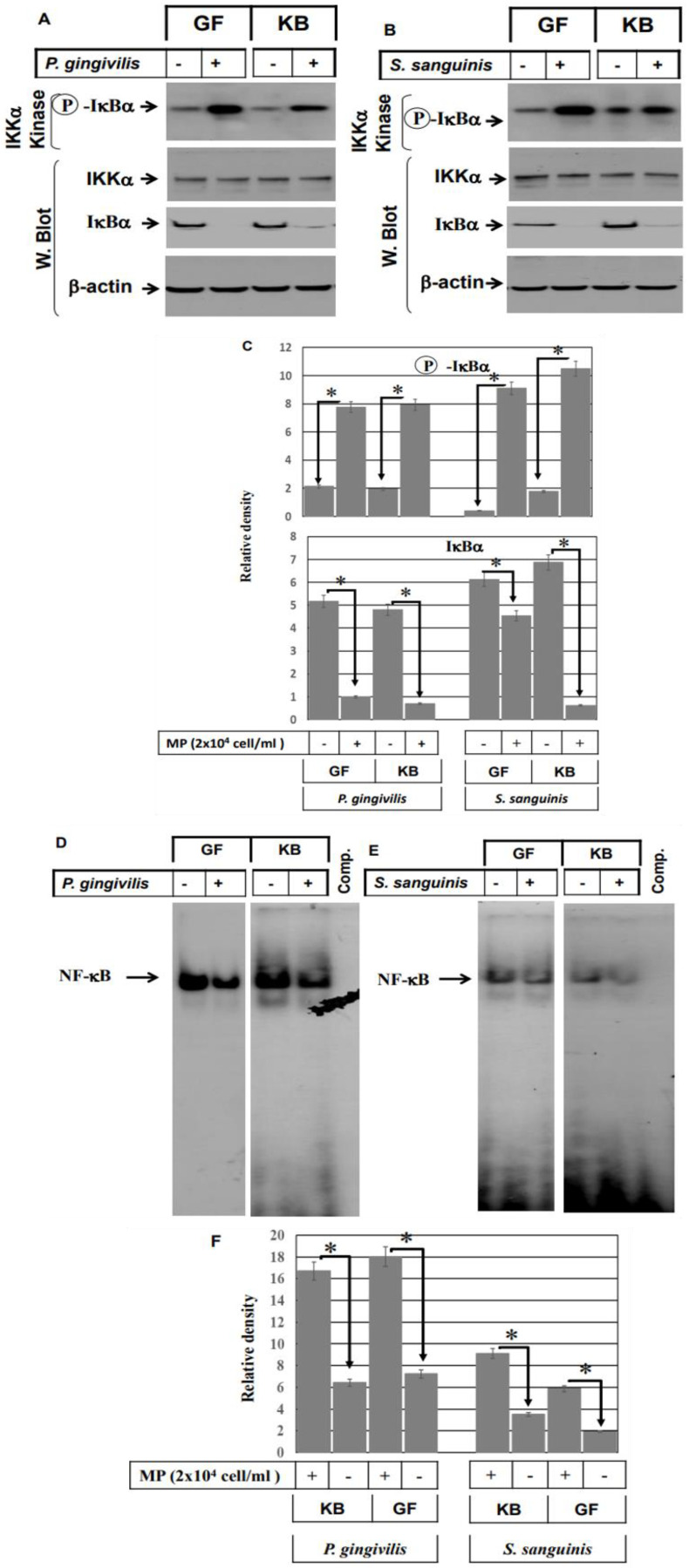
Induction of NF-κB pathway by both gram^−^ and gram^+^ bacterial pathogens in oral mucosa-derived cells. Both GF and KB cells were stimulated with either heat-inactivated *P. gingivalis* (2 × 10^4^ cells/mL) or heat-inactivated *S. sanguinis* (2 × 10^4^ cells/mL) microbial pathogens (MPs) for 48 h. Then, total protein lysate and nuclear extract were prepared to perform kinase assays, Western blot, and EMSA. Kinase assay demonstrates the activation of IKK by both in vitro phosphorylation of the IκBα substrate and NF-κB inhibition in both GF and KB cells by stimulation with either *P. gingivalis* (**A**) or *S. sanguinis*(**B**) when compared to control cells. Data are representative of three independent experiments performed separately. Western blot analysis of cytoplasmic protein of treated and control cells using anti-IκBα, IKKα, and actin antibodies. Western blot analysis of the protein stability of IκBα in the cytoplasmic protein of both cells demonstrates degradation of IκBα protein, evidence NF-κB activation in both GF and KB cells when stimulated with either heat-inactivated *P. gingivalis* (**A**) or heat-inactivated *S. sanguinis* (**B**) bacterial pathogens. IKKα and actin were used as internal controls for loading and transfer. (**C**) Analysis of band intensity on films is presented as the relative ratio of phospho-IκBα to the expression of IKK α and the degradation of IκBα to the expression level of actin. Bars represent the mean ± SD from three independent experiments performed separately, * *p* < 0.05. The DNA-binding activity of the transcription factors NF-κB was investigated by EMSA using nuclear extracts of treated and control cells. Stimulation of both GF and KB cells with heat-inactivated *P. gingivalis* (**D**) or heat-inactivated *S. sanguinis* (**E**) induced the DNA-binding activity of the transcription factor NF-κB when compared with those of control cells. (**F**) Analysis of band intensity of the transcription factor of NF-κB. Bars represent mean ± SD from three independent experiments performed separately, * *p* < 0.05.

**Figure 6 ijms-24-05348-f006:**
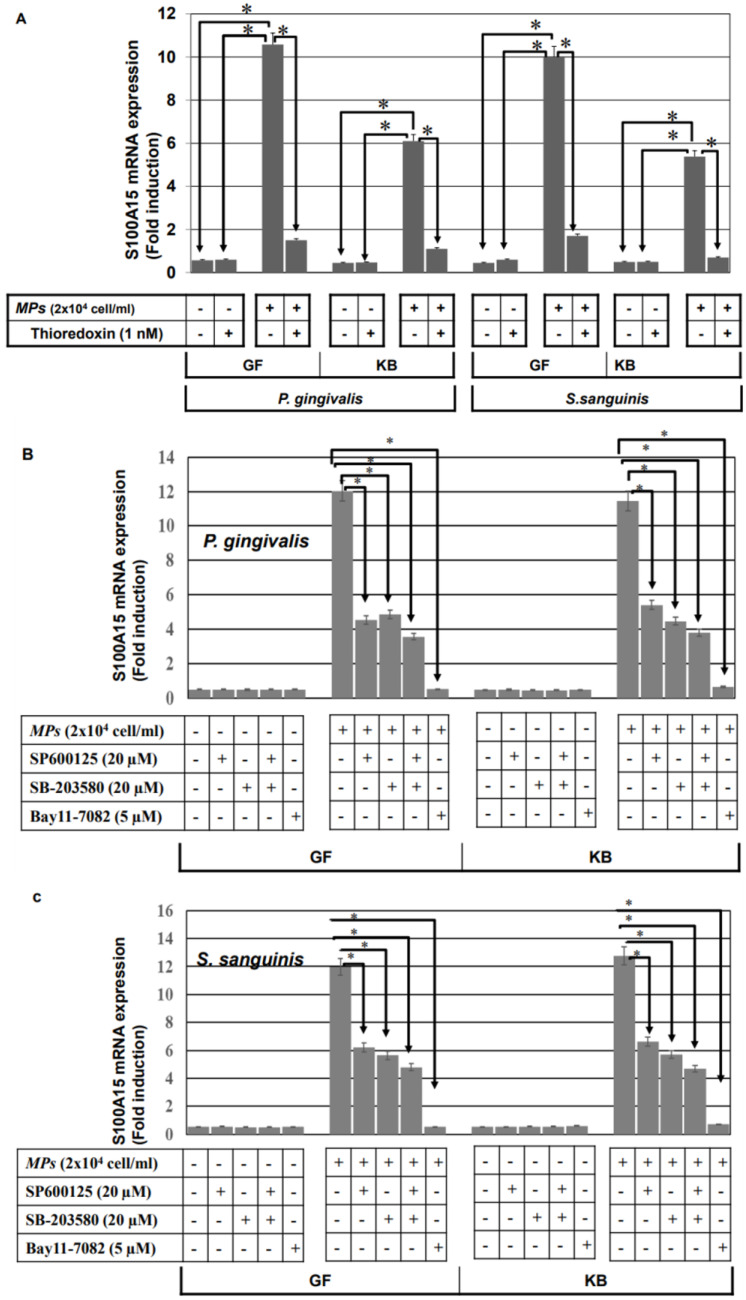
Bacterial pathogens-induced expression of S100A15 in oral mucosa-derived cells is mediated by the MAP kinase and NF-κB pathways. (**A**) GF and KB cells were pre-treated with Thioredoxin 1 h prior to stimulation with either heat-inactivated gram^−^
*P. gingivalis* or heat-inactivated gram^+^
*S. sanguinis* for 48 h. Total RNA was extracted from treated and control cells and subsequently subjected to qRT-PCR analysis to assess the relative transcription levels of S100A15. (**B**) GF and KB cells were pre-treated with inhibitors of JNK (SP600125), p38 (SB-203580), or NF-κB (Bay11-7082) 1 h prior to stimulation with either heat-inactivated *P. gingivalis* or heat-inactivated *S. sanguinis* for 48 h. Total RNA was extracted from treated and control cells and subsequently subjected to qRT-PCR analysis to assess the relative transcription levels of S100A15. (**C**) GF and KB cells were pre-treated with SP600125, SB-203580, or Bay11-7082 1 h prior to stimulation with heat-inactivated *P. gingivalis* or heat-inactivated *S. sanguinis* for 48 h. Total RNA was extracted from treated and control cells and subsequently subjected to qRT-PCR analysis to assess relative transcription levels of S100A15. Densitometric quantification of S100A15 mRNA over actin mRNA transcript. Each bar represents the mean ± SD of three independent experiments performed in duplicate, * *p* < 0.05.

**Figure 7 ijms-24-05348-f007:**
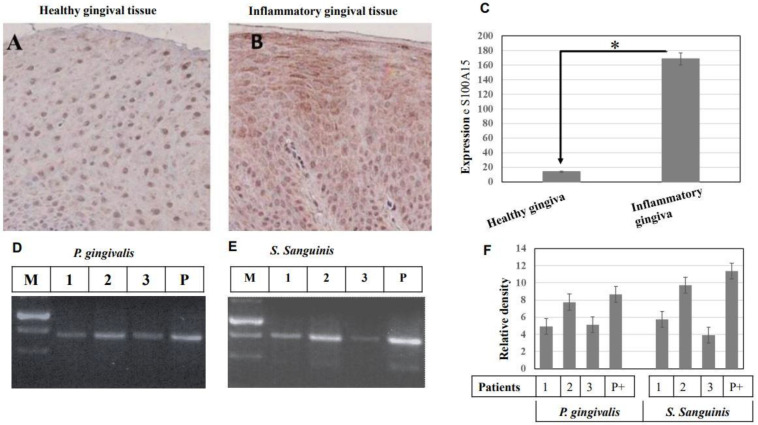
Immunohistochemical analysis of S100A15 protein in oral mucosa of patients with inflammatory gingiva. (**A**), Gingival tissues of healthy individuals (n = 10) were stained with anti-S100A15 antibodies. (**B**) Gingival tissues of patients with inflammatory gingiva (n = 10) were stained with anti-S100A15 antibody. Original magnification × 100. Scale bar indicates 100 μm. (**C**) Statistical Analysis of S100A15 expression in gingival tissues of patients with inflammatory gingiva (n = 10) as a relative ratio to the expression of S100A15 in gingival tissues of healthy individuals. Bars represent mean ± SD from inflammatory (n = 10) and healthy (n = 10) gingival tissues, * *p* < 0.05. (**D**,**E**) Detection of oral bacterial pathogens (gram^−^
*P. gingivalis* and gram^+^
*S. sanguinis*) in the saliva of patients with inflammatory gingiva by conventional PCR. Detection of both *P. gingivalis* and *S. sanguinis* in the saliva of patients with inflammatory gingiva by PCR. The PCR was performed using specific primers to amplify a 480 bp fragment of the 16S rRNA gene as a marker for *P. gingivalis* (**D**) and specific primers to amplify a 473 bp fragment of the UDP-N-acetylglucosamine-like protein gene as a marker for *S. sanguinis* (**E**). PCR products were separated on 2% agarose and stained with ethidium bromide. (**F**) Analysis of band intensity as the relative ratio of the amplified DNA fragment of 16S rRNA gene, the marker of *P. gingivalis,* and the amplified of the UDP-N-acetylglucosamine-the marker of *S. sanguinis* to the corresponding positive control. Bars represent the mean ± SD from three blots, * *p* < 0.05.

**Figure 8 ijms-24-05348-f008:**
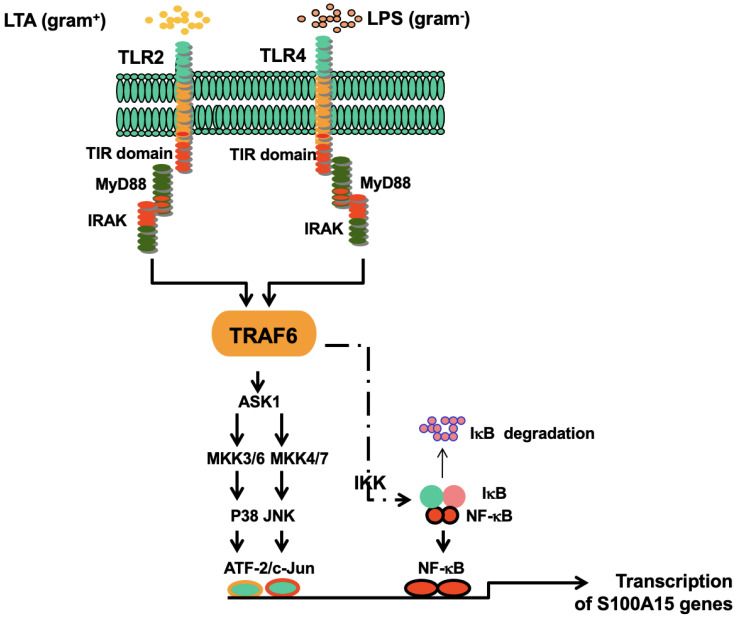
Proposed model for the mechanisms whereby the gram^−^ and gram^+^ bacterial pathogens induce the expression of the antimicrobial protein, S100A15, in the oral mucosa. The colonization of gram^−^ and gram^+^ bacterial pathogens result in the activation of both TLR4 and TLR2, respectively. As a consequence, TRAF6 plays a central role in mediating TLR4 and TLR2-induced signaling to NF-kB/ASK1-JNK-AP-1/ASK1-p38-ATF-2 pathways to enhance the transcriptional activation of S100A5, which subsequently functions as an antimicrobial protein to protect oral mucosa from bacterial pathogens.

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
