# Peer review of "Induction of Antimicrobial Protein S100A15 Expression by Oral Microbial Pathogens Is Toll-like Receptors-Dependent Activation of c-Jun-N-Terminal Kinase (JNK), p38, and NF-κB Pathways"

_ijms, 2023, doi:10.3390/ijms24065348_

Round 1

Reviewer 1 Report

The manuscript by Selimovic et al. presents experimental work investigating the molecular mechanisms of inducible expression of antimicrobial protein S100A15 in response to oral microbial pathogens. The work is well designed, methodologically adequate, the results are interesting and convincing and their interpretation is correct.

Although the experimental part of the paper is highly appreciated, the text of the manuscript is in need of comprehensive editing. Detailed comments are given below.

1) The supplementary material omits ELISA results for protein S100A15 contrary to the statement in the text (Line 107). Also due to this, the numbering of the supplementary figures in the text and in the supplementary material file does not match. The ELISA results should be presented or their mention should be removed from the text of the manuscript.

2) Although the manuscript is submitted to Int. J. Mol. Sci., a template for Microorganisms was used.

3) The characters of the Greek letters (α, κ) are not correctly displayed in many cases in the pdf file of the manuscript.

4) The captions for Figure 6B and 6C are the same. The figure itself consists of two parts only, A and B. Part A is not labelled accordingly.

5) Figure 7E does not look well in the pdf file of the manuscript, replace it with a higher quality version.

6) It seems that 'total protein lysate' is not quite correct term.

7) Add the abbreviation MPs and its meaning in the caption to Figure 2.

8) Add a reference to the results presented in Figure 2B to the text of the manuscript.

9) Line 110 – should be PCR, not Western blot.

10) Line 446 – You probably mean that all three proteins interact with the same mRNA, but you phrased it as if these proteins interacted with each other.

11) Line 478 – 600 seems to be a value of wavelength, not of optical density.

12) Describe more clearly the calculation of Fold induction (Lines 522-525), preferably adding a formula.

13) The text of the manuscript contains many misprints and sometimes even incomplete phrases (e.g. Lines 231-232). Here are a few examples of misprints, but these are only a small part of them, the whole text needs careful editing.

Line 33 – purifies (instead of ‘purified’)

Lines 239-240 – Total cell lysates was prepared and subjected or Western blot analysis (were/to)

Line 253 – of of

Line 440 – quit (should be ‘quite’)

Line 528 – washin

In the TMB and DAB names (Lines 563 and 649) 'benzidine' is misspelled in two different ways

Numerous mis-capitalizations: Both (Line 92), p. gingivalis (Line 186), S. Sanguinis (many times)

Author Response

Dear Editor,

Thank you very much for the encouraging comment regarding our Manuscript ID: ijms-2256835; “Induction of Antimicrobial Protein S100A15 Expression by Oral Microbial Pathogens is Toll-Like Receptors-dependent Activation of c-Jun-N-terminal kinase (JNK), p38, and NF-kB Pathways.”

As required, enclosed find please our response “Point-for-Point” to the valuable comments of Reviewer 1. Thank you very much for your consideration.

On behalf of all my coauthors

Reviewer 2 Report

I have read the manuscript entitled “Induction of Antimicrobial Protein S100A15 Expression by Oral Microbial Pathogens is Toll-Like Receptors-dependent Activation of c-Jun-N-terminal kinase (JNK), p38, and NF-kB Pathways” with great interest and I think it is in principle suited for a publication in the IJMS, Section “Biochemistry”. The authors demonstrated that S100A15 is induced in oral mucosa derived cell lines by bacterial pathogens, and they provided insight into the mechanisms of the induction and the transcription regulation of S100A15 by ASK1-JNK-AP1, ASK1-p38-ATF-2 and NF-kB pathways.

The manuscript brings some new information to the scientific community. However, I also have concerns and comments.

Comments:

Lines 59-62. “S100 proteins are also involved in the modulation of a wide variety of intra- and extracellular functions, including cytokine-like and chemokine-like modulation of advanced glycation end products (RAGE) and Toll-like receptor 4 (TLR4) dependent signalling cascades [12].” I think that S100 proteins are not discussed in the cited work #12. Please check and if necessary, replace with a more suitable reference (e.g., https://www.ncbi.nlm.nih.gov/pmc/articles/PMC8121140/ ).

Lines 71-73. “These antimicrobial peptides have a broad spectrum of activity against both Gram-negative and Gram-positive bacteria as well as yeast and viruses [15-20]. I think that antimicrobial effects of antimicrobial peptides are not discussed in the cited work #15. Please check.

Lines 162-163. “Densitometric quantification of S100A15 mRNA over actin mRNA transcript.” I think that this procedure needs to be described in the “Materials and Methods” section.

Lines 181-183. “The role of LPS-induced ASK1 activity in the activation of mitogen-activated protein kinases including, c-Jun N-terminal kinase (JNK) and p38 pathways has been reported in several studies [27, 28].” I think that ASK1 is not discussed in the cited work #27. At the same time LPS is not discussed in the cited article #28. 

Figure 3. The quality of the image should be increased.

Figure 5. Please check the designation of the panels D and E.

Figure 6. Please check the designations of the panels A and C.

Figure 7. Please check the contrast and/or quality of the panel E

Lines 429-432. “The role of ASK1 in the regulation of MAP kinase kinase 4 (MKK4)-JNK and MAP kinase kinase 7 (MKK7)-JNK, MAP kinase kinase 3 (MKK3)-p38 and MAP kinase kinase 6 (MKK6)-p38 pathways has been widely documented in several studies [33-35].” I think that MKK4, MKK7, MKK3, and MKK6 are not discussed in the cited article #34. Please check.

Line 471. “] has been added to the text of the manuscript.” Please check.

Lines 496-497, 507. Analysis of S100A15 by qRT-PCR was performed as previously described [21]”. It is repeat.

Lines 570, 583. “EMSAs were performed as described [22].” “In vitro kinase assay was performed as described previously [22].” I think that the methods are not described in the cited work #22. This article (#22) contains links to earlier works by the authors (in the corresponding “Materials and methods” section). 

Lines 636-637. “The expression of S100A15 in gingival tissues was detected by immunohistochemistry using a S100A15 antibody according to the standard methods [23].” I think that “standard methods using S100A15 antibody” are not described in the cited article. Please check.

Best regards.

Author Response

Dear Editor,

Thank you very much for the encouraging comment regarding our Manuscript ID: ijms-2256835; “Induction of Antimicrobial Protein S100A15 Expression by Oral Microbial Pathogens is Toll-Like Receptors-dependent Activation of c-Jun-N-terminal kinase (JNK), p38, and NF-kB Pathways.”

As required, enclosed find please our response “Point-for-Point” to the valuable comments of Reviewer 2. Thank you very much for your consideration.

On behalf of all my coauthors
